# Treatment of Textile Wastewater Using Advanced Oxidation Processes—A Critical Review

Yiqing Zhang [1], Kashif Shaad [1,*], Derek Vollmer [1] and Chi Ma [2]

1   Moore Center for Science, Conservation International, Arlington, VA 22202, USA;
    yizhang@conservation.org (Y.Z.); dvollmer@conservation.org (D.V.)
2   Sateri, Jiujiang 332017, China; chi_ma@sateri.com.cn
*   Correspondence: kshaad@conservation.org

**Abstract:** Textile manufacturing is a multi-stage operation process that produces significant amounts of highly toxic wastewater. Given the size of the global textile market and its environmental impact, the development of effective, economical, and easy-to handle alternative treatment technologies for textile wastewater is of significant interest. Based on the analysis of peer-reviewed publications over the last two decades, this paper provides a comprehensive review of advanced oxidation processes (AOPs) on textile wastewater treatment, including their performances, mechanisms, advantages, disadvantages, influencing factors, and electrical energy per order ($E_{EO}$) requirements. Fenton-based AOPs show the lowest median $E_{EO}$ value of 0.98 kWh m$^{-3}$ order$^{-1}$, followed by photochemical (3.20 kWh m$^{-3}$ order$^{-1}$), ozonation (3.34 kWh m$^{-3}$ order$^{-1}$), electrochemical (29.5 kWh m$^{-3}$ order$^{-1}$), photocatalysis (91 kWh m$^{-3}$ order$^{-1}$), and ultrasound (971.45 kWh m$^{-3}$ order$^{-1}$). The Fenton process can treat textile effluent at the lowest possible cost due to the minimal energy input and low reagent cost, while Ultrasound-based AOPs show the lowest electrical efficiency due to the high energy consumption. Further, to explore the applicability of these methods, available results from a full-scale implementation of the enhanced Fenton technology at a textile mill wastewater treatment plant (WWTP) are discussed. The WWTP operates at an estimated cost of CNY ¥1.62 m$^{-3}$ (USD \$0.23 m$^{-3}$) with effluent meeting the China Grade I-A pollutant discharge standard for municipal WWTPs, indicating that the enhanced Fenton technology is efficient and cost-effective in industrial treatment for textile effluent.

**Keywords:** textile wastewater; advanced oxidation process; electrical energy per order; Fenton process

## 1. Introduction

Textile manufacturing is an important pillar of many economies, particularly in China, India, Vietnam, Bangladesh, and Pakistan [1–3]. A technologically complex industrial chain utilizes a wide array of fibers from natural sources (such as cotton, flax, silk, and wool) to synthetic/man-made sources (such as viscose, nylon, polyester, and acrylic) and converts the raw materials into useful finished products [4]. This industrial chain includes a series of multi-step processes, such as spinning, weaving, knitting, de-sizing, sizing, scouring, washing, bleaching, mercerizing, dyeing, printing, and finishing [4,5].

Textile manufacturing produces significant amounts of wastewater as a result of the water and chemicals consumed during each step. As of 2017, it is estimated that more than 700,000 tons of toxic dyeing wastewater is generated annually, with nearly 200,000 tons being discharged as effluent without proper treatment [6,7]. The World Bank estimated that textile dyeing and finishing wastewater accounts for nearly 17–20% of total industrial wastewater [8]. The chemical reagents used in the textile process are synthesized in a variety of ways and have complex molecular structures, ranging from compounds of low molecular weight to polymers [9]. Due to the large volume of effluents and the great variety of chemical composition, textile wastewater poses serious threats to downstream ecosystems and human health if discharged improperly [10]. The highly colored component

can impede passage of sunlight, inhibit the photosynthesis of aquatic plants, and affect the reoxygenation and self-purification of water bodies, thereby acting as a source of eutrophication [11]. In addition, textile dyes are frequently found to be toxic, mutagenic, and carcinogenic, which may cause human diseases such as skin irritation, headache, nausea, respiratory ailments, and congenital malformation [1,12].

Numerous articles on the treatment of textile wastewater using conventional technologies have been published, indicating that the effectiveness of conventional treatment processes in remediating these chemically stable and/or biologically recalcitrant compounds falls short of requirements [7,10,13,14]. Membrane filtration has a significant potential for the reclamation of dye effluents and a high resistance to adverse chemical environments. However, its application is restricted due to membrane fouling and high initial investment and operation costs [2,15,16]. Adsorption technology is a low-cost method for removing a wide variety of dyes by collecting or transferring them from liquid phase to solid surface. However, it is only capable of removing a portion of contaminants and requires a long contact time [17]. Coagulation/flocculation is an effective pretreatment method for removing disperse and sulfur dyes at a low initial cost. However, it is inefficient against highly soluble, azo, reactive, acid, and basic dyes, and generates a significant amount of sludge as a result of coagulant consumption [13,18]. Biological treatment technology is cost-competitive and effective in removing direct, disperse, and basic dyes, with a particular emphasis on biochemical oxygen demand and suspended solids removal. However, it requires a long hydraulic retention time and correspondingly large space and it shows poor applicability at large-scale due to the presence of bio-persistent dyes and toxic substances [19–21].

As a set of alternative treatment procedures to these conventional methods, advanced oxidation processes (AOPs) have received increasing attention due to the in situ generation of highly reactive species including hydroxyl radicals (HO$^\bullet$) under near ambient temperature and pressure [22]. HO$^\bullet$ is a highly reactive oxidizing agent with a high oxidation potential of 2.15–2.8 V, making it the second most reactive oxidant known (Table S1). It can attack recalcitrant pollutant species in water rapidly and non-selectively through hydrogen abstraction, electron transfer, and radical addition [23]. The reaction rate constants between HO$^\bullet$ and various organic compounds range in the order of $10^6$–$10^9$ M$^{-1}$ s$^{-1}$ [24]. In addition, HO$^\bullet$ has a short lifetime of several nanoseconds in water, leading to its self-elimination from the treatment system [25].

Many bench-scaled laboratory studies have reported that textile wastewater can be effectively treated by different AOPs, such as photochemical [26,27], photocatalysis [28,29], Fenton [20,30], electrochemical [31,32], and ultrasound [33,34] (Figure 1). However, the majority of the studies focused exclusively on one or two AOPs, with only a few comparing the effectiveness of various AOPs in textile wastewater treatment. In addition, AOPs exhibit different operational costs (including electrical energy and chemical reagents) that must be considered when applying at an industrial scale. Miklos evaluated the energy efficiency of emerging AOPs for water and wastewater treatment and found significant cost differences between various types of AOPs [35]. Mahamuni and Adewuyi estimated the economic cost of AOPs for wastewater treatment that incorporate ultrasound [36]. Despite the increasing trend of publications on the AOPs' operation costs during the last two decades (Figure 2), to the best of our knowledge, no study has compared the costs of textile wastewater treatment using various AOPs. Additionally, promising results from laboratory experiments have sparked interest in industrial-scale implementation of AOPs. Paździor et al. used two textile factories in Poland as industrial case studies to demonstrate the potential benefits of using AOP-biological method for textile wastewater treatment [4]. However, as only a few case studies of full-scale textile wastewater treatment plants with AOP have been reported so far, there is an information gap for the textile companies and regulatory agencies looking to adopt improved treatment methods.

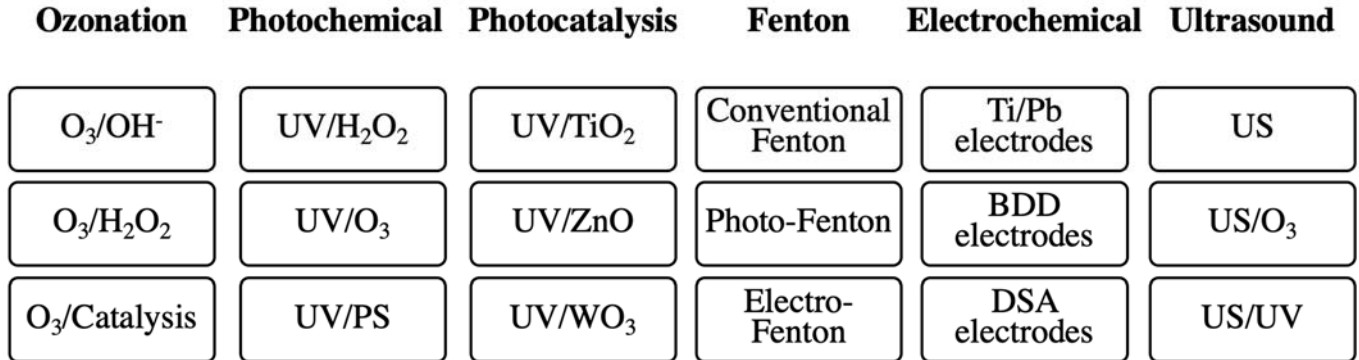

**Figure 1.** Advanced oxidation processes used in textile wastewater treatment.

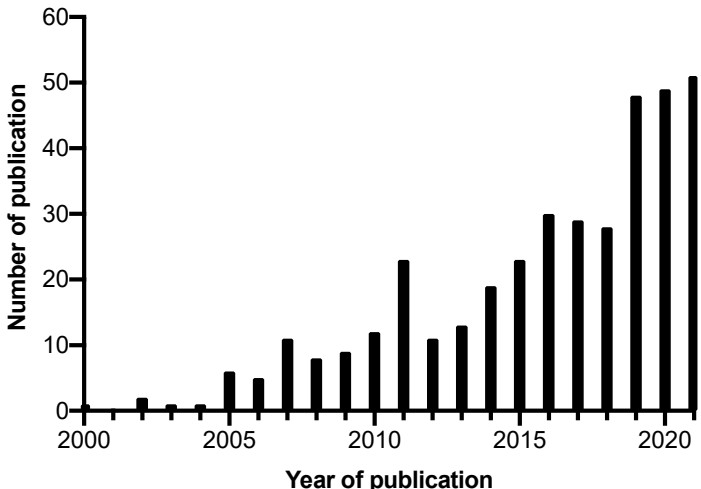

**Figure 2.** Number of publications on the cost assessment of AOPs for textile wastewater treatment during the last two decades (source: Advanced search in Scopus; accessed on 30 November 2021; keywords: textile wastewater AND advanced oxidation process AND water AND electrical energy per order, Link: https://www.scopus.com/results/results.uri?sort=plf-f&src=s&sid=70 b17aeb0a730bfe6b65889335649869&sot=a&sdt=a&sl=91&s=textile+wastewater+AND+advanced+ oxidation+process+AND+water+AND+electrical+energy+per+order&origin=searchadvanced& editSaveSearch=&txGid=fe743bafe94155f43d8eed171b718d77).

    The primary objective of this paper is to assess the state-of-the-art in terms of applying AOPs to textile wastewater, and is based on a review of over 200 peer-reviewed articles published over the last two decades (from 1999 to 2020). The remainder of this review article is divided into three parts. The first part (Section 2) discusses the characteristics of raw textile wastewater in general and compares it with the related discharge standard in China. The second part presents our systematic review of the effectiveness of AOPs using scaled-down laboratory experiments and provides a comprehensive overview of the types, major mechanisms, operating parameters, and advantages and disadvantages of each class of AOP (Section 3). Following this, a discussion on the practical implications of energy costs associated with the various methods (Section 4). The third part (Section 5) provides a concise overview of a full-scale implementation of an AOP-based textile mill wastewater treatment plant in China, including the schematics of the WWTP and a brief discussion of the reported results from treatment. Finally, in Section 6, we conclude the article by presenting some key unsolved issues and include suggestions for future research and development. By compiling information on the technical feasibility and economic cost of AOPs for textile wastewater treatment and introducing a real application case study, this

paper can be regarded as a valuable resource for AOP researchers, textile companies, and regulatory agencies worldwide.

## 2. Raw Textile Wastewater and Discharge Standard

Chemical oxygen demand (COD), biological oxygen demand ($BOD_5$), and suspended solids (SS) concentration in untreated or inadequately treated textile wastewater are found to be extremely high. Table 1 summarizes the characteristics of raw textile wastewater as reported by recent studies. Based on these reported figures, textile wastewater is seen to have a COD concentration that varies between approximately 413–8000 mg $L^{-1}$, $BOD_5$ concentration between 160–491 mg $L^{-1}$, SS concentration between 64–2545 mg $L^{-1}$, ADMI unit color between 625–2175, and a basic solution pH [37–54].

**Table 1.** Characteristics of raw textile wastewater from recent studies and discharge standards of major pollutants for municipal wastewater treatment plants in China.

| | | COD (mg $L^{-1}$) | $BOD_5$ (mg $L^{-1}$) | SS (mg $L^{-1}$) | Color (ADMI Unit) | pH | Reference |
|---|---|---|---|---|---|---|---|
| Raw textile wastewater | | 729 | | 558 | | 7.11 | [37] |
| | | 550 | | 887 | | 8.29 | [38] |
| | | 1476 | 491 | | | 10.66 | [39] |
| | | 5800 | 181 | 2000 | | 9 | [40] |
| | | 3422 | | 1112 | | 6.95 | [41] |
| | | 1600–1900 | | | 1500 | 12.5 | [42] |
| | | 1000 | 300 | 2545 | | 9.0 | [43] |
| | | $2300 \pm 400$ | | $300 \pm 100$ | | $13 \pm 1$ | [44] |
| | | $810 \pm 50.4$ | $188 \pm 15.2$ | $64 \pm 8.5$ | | 7.8 | [45] |
| | | $1132.6 \pm 2.5$ | | $1697 \pm 7$ | | $7.7 \pm 0.115$ | [20] |
| | | 3422 | | 1112 | | 6.95 | [46] |
| | | 1658–1692 | | | | 7.32–7.94 | [47] |
| | | 838 | 218 | 200 | 1300 | 6.3 | [48] |
| | | 3828 | 433 | 416 | | 12.5 | [49] |
| | | 750 | 160 | | | 9 | [50] |
| | | 1150 | 170 | 150 | | 10 | [51] |
| | | 314–404 | | | 609–975 | 9.43–9.58 | [52] |
| | | 1354 | | 84.2 | 2175 | 8.84 | [53] |
| | | 413 | | 289 | 625 | 8.7 | [54] |
| China municipal WWTP discharge standard | Level I-A | 50 | 10 | 10 | 30 | 6–9 | [55] |
| | Level I-B | 60 | 20 | 20 | 30 | 6–9 | |
| | Level II | 100 | 30 | 30 | 40 | 6–9 | |
| | Level III | 120 | 60 | 50 | 50 | 6–9 | |
| China textile WWTP discharge standard | Direct discharge | 80 | 20 | 50 | 50 | 6–9 | [56] |
| | Indirect discharge | 200 | 50 | 100 | 80 | 6–9 | |

To mitigate the harm caused by this wastewater to human and environmental health, many countries have legalized stringent discharge standards for textile wastewater treatment plants (WWTP). An example is the national standard adopted by China (PRC National Standard GB 18918–2002 outlined in Table 1) in 2002 [55]. This sets a "Maximum discharge standard of pollutants for municipal wastewater treatment plants". The criteria under the national standard are divided into four levels with Level I-A having the highest requirement. Level I-A should be met when WWTP effluents are discharged into rivers or lakes that have limited dilution ability or are used as municipal landscape water or general reuse water. Another discharge standard was established in 2012 to regulate the effluent from the textile dyeing and finishing industry, covering discharge limits and requirements for monitoring and controlling water pollutants in the textile dyeing and finishing industry or production facility [56]. The noticeable gap between the pollutant concentration in raw

textile wastewater and discharge standards of major pollutants for WWTP indicate the necessity of effective treatment methods (Figure 3).

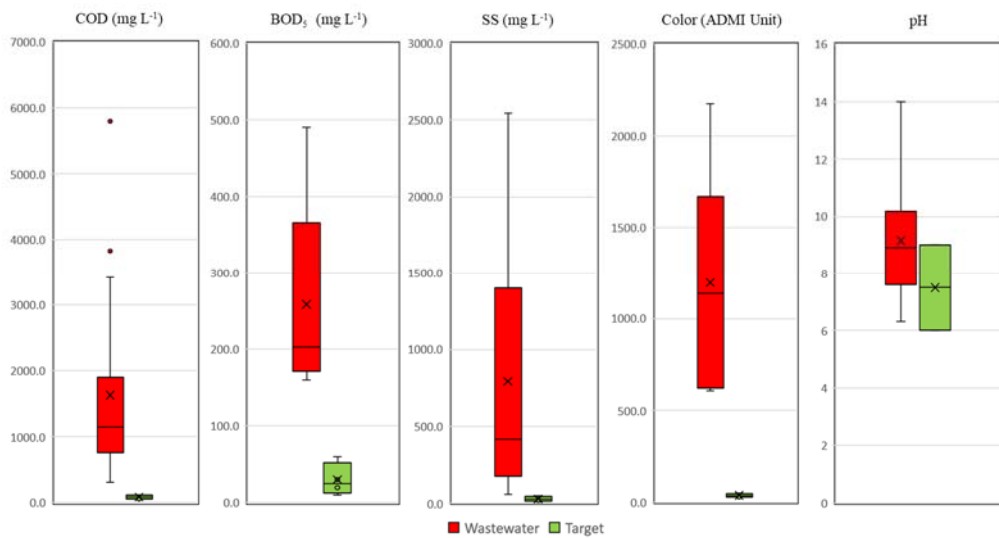

**Figure 3.** Gap between the pollutant concentration in raw textile wastewater and discharge standards of major pollutants.

## 3. Types of AOP and Comparison

AOPs have been widely investigated as innovative approaches to treat textile wastewater. Based on the different ways used to generate oxidation agents, AOPs can be classified into six categories: ozonation, photochemical, photocatalysis, Fenton, electrochemical, and ultrasound. Their mechanisms, advantages, and disadvantages are summarized in Table 2. The following sub-sections go into detail about their characters and operating parameters.

**Table 2.** The mechanisms, advantages, and disadvantages of various AOPs.

| Process | Mechanisms | Pros | Cons |
|---|---|---|---|
| Ozonation | (1) $O_3$ at elevated pH:<br>$3O_3 + OH^- + H^+ \rightarrow 2HO^\bullet + 4O_2$<br>(2) $O_3/H_2O_2$:<br>$2O_3 + H_2O_2 \rightarrow 3O_2 + 2HO^\bullet$ | (1) High decolorization<br>(2) High biodegradability | (1) Low mass transfer efficiency<br>(2) Potential toxic by-products<br>(3) Gaseous $O_3$ in the off-gas has to be removed |
| Photochemical | (1) $UV/H_2O_2$:<br>$H_2O_2 + h\nu \rightarrow 2HO^\bullet$ ($\Phi = 1.0$)<br>(2) $UV/O_3$:<br>$O_3 + H_2O + h\nu \rightarrow 2HO^\bullet + O_2$<br>(3) $UV/PS$:<br>$S_2O_8^{2-} + h\nu \rightarrow 2SO_4^{\bullet-}$ ($\Phi = 1.8$)<br>(4) $UV/PMS$:<br>$HSO_5^- + h\nu \rightarrow HO^\bullet + SO_4^{\bullet-}$<br>($\Phi = 1.04$ at pH = 7) | (1) Simple and easy operation<br>(2) Clean and no sludge production<br>(3) Relatively commercial oxidants<br>(4) Disinfect water | (1) Waters with low UV light transmittance cannot be treated<br>(2) Low quantum yield of oxidants |
| Photocatalysis | $S + h\nu \rightarrow S (e^- + h^+)$<br>$S(h^+) + H_2O \rightarrow H^+ + HO^\bullet$<br>$S(e^-) + O_2 \rightarrow O_2^{\bullet-}$<br>$O_2^{\bullet-} + H^+ \rightarrow HO_2^\bullet$<br>$HO_2^\bullet + S(e^-) + H^+ \rightarrow H_2O_2$<br>$H_2O_2 + S(e^-) \rightarrow OH^- + HO^\bullet$ | (1) Mild operation conditions<br>(2) Photochemically stable and non-toxic catalysts<br>(3) Potential to use sunlight as clean and economic photo-source | (1) Low light use efficiency and low quantum efficiency of $HO^\bullet$ generation<br>(2) Mass transfer limitations to the surface of the immobilized catalyst on a substrate;<br>(3) High recombination rate for the photoelectron and holes pairs |

**Table 2.** *Cont.*

| Process | Mechanisms | Pros | Cons |
|---|---|---|---|
| Fenton | (1) Classical Fenton:<br>$Fe^{2+} + H_2O_2 \rightarrow Fe^{3+} + OH^- + HO^\bullet$<br>$(k = 70\ M^{-1}s^{-1})$<br>$Fe^{3+} + H_2O_2 \rightarrow Fe^{2+} + HO_2^\bullet + H^+$<br>$(k = 0.001\text{–}0.01\ M^{-1}s^{-1})$<br>(2) Photo-Fenton:<br>$H_2O_2 + h\nu \rightarrow 2HO^\bullet$<br>$Fe(OH)^{2+} + h\nu \rightarrow Fe^{2+} + HO^\bullet$<br>$Fe^{3+} + H_2O_2 + h\nu \rightarrow Fe^{2+} + HO^\bullet + H^+$<br>(3) Electro-Fenton:<br>$O_2 + 2H^+ + 2e^- \rightarrow H_2O_2$<br>$Fe^{3+} + e^- \rightarrow Fe^{2+}$ | (1) No energy input requirement (for classical Fenton process)<br>(2) Relatively inexpensive chemicals<br>(3) Simple and flexible operation | (1) Narrow pH range<br>(2) Large amount of sludge production<br>(3) High concentration of $Fe^{2+}$ in the effluent |
| Electrochemical | $M + H_2O \rightarrow M(HO^\bullet) + H^+ + e^-$<br>$R + M(HO^\bullet) \rightarrow M + mCO_2 + nH_2O + pX$ | (1) Environmental compatibility<br>(2) Possibility of automation<br>(3) No chemical required and no sludge produced | (1) Limited mass transfer efficiency<br>(2) High requirement of electricity<br>(3) Potential poisoning effect |
| Ultrasound | (1) US:<br>$H_2O +\ ))) \rightarrow HO^\bullet + H^\bullet$<br>$H^\bullet + O_2 +\ ))) \rightarrow HOO^\bullet$<br>where ))) refers to the US irradiation.<br>(2) US/$O_3$:<br>$O_3 +\ ))) \rightarrow O_2(g) + O(^3P)(g)$<br>$O(^3P)(g) + H_2O(g) \rightarrow 2HO^\bullet(g)$ | (1) No chemical required and no sludge produced<br>(2) Improves the effect of chemical reaction | (1) Energy-intensive<br>(2) A large amount of dissolved oxygen required |

### 3.1. $O_3$-Based AOP

### 3.1.1. Description

Ozone ($O_3$) has been frequently used to disinfect microorganisms and oxidize micropollutants since the late 19th century [57]. Ozonation can degrade textile dyes in two distinct ways depending on the solution pH, i.e., through direct attack by molecular $O_3$ or through indirect reaction with generated radical species. Under acidic conditions, $O_3$ can act as an electrophile on specific functional groups in textile dye compounds (also referred to as ozonolysis). With a high oxidation potential of 2.07 V, $O_3$ can remove a wide range of organic compounds from water and wastewater effectively. Specific functional groups of the textile dye can be attacked via electrophilic interactions, mainly electron-rich functional groups, such as double bonds, amines, and aromatic rings, resulting in the generation of aldehydes, carboxylic acids, and other by-products [27,58,59]. At basic pH, $O_3$ rapidly reacts with $OH^-$ to yield $HO^\bullet$ and other radical species by a series of complex chain reactions [22,60].

$O_3$ has also been coupled with powerful oxidants or catalysts. $O_3/H_2O_2$ has been extensively investigated as a single process or pre-treatment step for dye removal from textile wastewater [61–64]. The addition of $H_2O_2$ to the ozonation process (also known as peroxonation) can lead to the decomposition of $O_3$ and the generation of highly reactive $HO^\bullet$. Catalysts containing $O_3$ can also be used to increase the ozonation reaction rates in textile wastewater treatment [59]. Catalytic ozonation can be classified into homogeneous and heterogeneous. Homogenous catalytic ozonation decomposes ozone using transition metal ions, such as $Fe^{2+}$ [48,65], $Fe^{3+}$ [64], zero-valent iron nanoparticles (nZVI) [48], and $Mn^{2+}$ [66], etc. Heterogeneous catalytic ozonation uses metal oxides or other solid materials, such as $Al_2O_3$ [67], $MnO_2$ [66,68], $Ca(OH)_2$ [69], granular activated carbon (GAC) [70], and perfluorooctanoic acid (PFOA) [67], etc.

The main advantages of $O_3$-based AOPs are their high decolorization and high biodegradability. $O_3$ is effective in decolorizing dye wastewaters since it tends to at-

tack conjugated double bonds that are frequently associated with color [71]. Bahan et al. investigated the ozonation of biologically treated woolen textile effluent and found that the decolorization efficiency reached around 98–99% after 40 min of ozonation with a corresponding ozone absorption rate of 58 mg $L^{-1}$ [26]. Catalytic ozonation processes have historically been reported to be more efficient than ozonation alone, since oxidants or catalysts can accelerate the decomposition rate of $O_3$ in water, generating a greater amount of reactive $HO^\bullet$. The biodegradability (characterized by $BOD_5/COD$) of textile wastewater can be significantly increased after treatment with $O_3$-based AOPs. The reason for this is that $O_3$ has a proclivity for directly attacking refractory organic pollutants and forming biodegradable intermediate products with a low molecular weight [72]. Turhan and Ozturkcan used ozonation to evaluate the biodegradability of aqueous Reactive Orange 16 solutions [27]. They found that the $BOD_5/COD$ ratio of the dye solution increased from 22.6 to 64.9 after 30 min of ozonation. Churchley et al. found that the non-biodegradable dye waste was converted to biodegradable forms after ozonation, resulting in an increase in the biotic score of the receiving river [73]. Malik et al. demonstrated that pretreatment with nano-catalytic ozone increased the $BOD_5/COD$ ratio to 0.61 (134.6%) along with COD, color, and toxicity removal up to 73.5%, 87%, and 92%, respectively [48].

Industrial application of $O_3$-based AOPs has been limited by their low mass transfer efficiency, potential toxic by-products, and susceptibility to pH changes. The total reaction rate of the ozonation process can be affected by mass transfer from gas to liquid. The low gas-liquid mass transfer rate is the rate-limiting step of ozonation, since ozone has low solubility and is easily lost in the off-gas exiting the reactor [74]. This subsequently leads to a low utilization rate and high operation costs. A high ozone dose must be applied to meet the discharge standards, which makes this process energy-intensive [75]. When $O_3$ reacts with organic compounds in water, toxic and carcinogenic by-products may be produced, such as formaldehyde, ketones, phenols, nitromethanes, bromates, and N-nitrodimethylamine [76]. Wang et al. evaluated the ozonation-induced toxicity of C.I. Remazol Black 5 and its degradation products [77]. The results showed that the first by-product after short-term ozonation possessed a high potential for toxicity as determined by a bioluminescence test (*Vibrio fischeri*) and a neutral red cytotoxicity assay test (rat hepatoma cells). The toxic intermediates (in the study) decreased after long-term ozonation treatment due to the significant increase in microbial biodegradability.

### 3.1.2. Influencing Parameters

Various operating parameters can influence the treatment efficiency of $O_3$-based AOPs, including pH, $O_3$ dose, and catalyst concentration [78]. Solution pH is a critical parameter that significantly influences direct ozonation efficiency due to the two distinct ozonation mechanisms. Alkaline pH conditions favor ozonation efficiency because much less $HO^\bullet$ is generated in acidic conditions. Turhan et al. found that the COD removal of direct dyestuff in wastewater increased from 23.33 to 64.96% when pH increased from pH 6.5 to 12 after 2 h of ozone bubbling [71]. Muthukumar et al. evaluated the efficiency of ozone treatment for Acid Red 88 at pH values ranging from 3 to 11 and reported that the maximum COD removal of 64% was obtained at alkaline pH [79]. The pH of the solution is also important for the $O_3/H_2O_2$ reaction. At acidic pH values, $H_2O_2$ reacts very slowly with $O_3$. However, at pH values greater than 5, the rate of $O_3$ decomposition by $H_2O_2$ increases significantly. At pH values around 11.6, even a trace amount of $H_2O_2$ is more effective in initiating the decomposition of $O_3$ than the $OH^-$ ion. However, the benefits of using $O_3/H_2O_2$ in textile wastewater at high pH values are limited due to strong competition reactions with $OH^-$ and already efficient radical formation with $O_3$. Arslan et al. reported that adding 0.1–10 mM $H_2O_2$ had no effect on the rate of dyestuffs decolorization when compared to $O_3$ treatment alone [61]. At pH 11.5 with optimum concentration of $H_2O_2$, the highest decolorization rate was still lower than $O_3$ alone at alkaline pH.

Ozone dose has a significant effect on the removal efficiency of textile dyes. The mass transfer rate of ozone improves with the increasing ozone dose. Increased ozone absorption

and reaction with pollutant molecules results in an increase in the decomposition of textile dyes [78]. However, a higher $O_3$ dose cannot be considered optimal since it increases the $O_3$ consumption per unit dye solution. The efficiency of $O_3$ utilization depends on the batch reactor. The fraction of unused $O_3$ is approximately 0.5 in most of the experiments, as reported by Tehrani-Bagha et al. [80].

Catalyst concentration also significantly influences removal efficiency of pollutants. For $O_3$/$H_2O_2$ process, the $H_2O_2$ dose can significantly increase the reaction rate of $O_3$/$H_2O_2$ at low levels. However, no additional treatment effect can be observed when the $H_2O_2$ dose exceeds the optimum $H_2O_2$ concentration. The excess $H_2O_2$ may act as radical scavengers and inhibit the oxidation reaction by generating less reactive species. Khadhraoui et al. reported that the color removal efficiency of Congo Red was lower after $O_3$/$H_2O_2$ treatment as compared to ozone treatment alone [63]. The reason is due to the scavenging effect of the high $H_2O_2$ concentration used. The actual optimum $[H_2O_2]/[O_3]$ molar ratio is always higher than the stoichiometric optimum of 0.35. Additional $H_2O_2$ is required due to the low absorption rate of $O_3$ by target compounds and the existence of radical scavengers in real water matrices. Arslan et al. evaluated the degradation effect of dyestuffs by $O_3$/$H_2O_2$ using $H_2O_2$ concentrations up to 10 mM [61]. The optimal $H_2O_2$ concentration was around 1 mM to achieve the highest decolorization rate and DOC removal at neutral pH. The optimal molar ratio of $[H_2O_2]/[O_3]$ was 1.45. Additional recent publications on the textile wastewater treatment using $O_3$-based AOPs are listed in Table S2.

*3.2. Photochemical*

3.2.1. Description

Homogenous photochemical reactions are characterized by the electronic excitation of chemical molecules and are initiated by the absorption of energy in the form of light [81]. The most commonly applied photolysis radiation has a wavelength of 200–400 nm, which is in the UV spectrum [82]. The photoactive textile dye molecules can directly absorb luminous radiation to reach an excited state, leading to the bond rupture and, thus, degradation. UV light can also react with powerful oxidizing chemicals to generate radicals through a photochemical reaction. The radicals show high removal efficiencies for textile dyes with low UV absorption [83].

The UV/$H_2O_2$ process has been the most commercially applied photochemical process. One mole $H_2O_2$ can be photolyzed to produce two moles $HO^\bullet$. However, due to a low molar absorption coefficient of $H_2O_2$, it is only used in trace amounts during UV irradiation, limiting the UV/$H_2O_2$ process [84]. $O_3$ absorbs more UV light than $H_2O_2$ at the same dosage. The molar absorption coefficient of $O_3$ (3600 $M^{-1}$ $cm^{-1}$) is 200 times higher than that of $H_2O_2$ (18.6 $M^{-1}$ $cm^{-1}$) at a UV wavelength of 254 nm, suggesting $O_3$ photolysis should be more efficient than $H_2O_2$ photolysis [85]. Therefore, the UV/$O_3$ process and a further combined UV/$H_2O_2$/$O_3$ process have been investigated in textile wastewater treatment [86]. The coupling of persulfate (PS, $S_2O_8^{2-}$) and peroxymonosulfate (PMS, $HSO_5^{-}$) with UV irradiance can significantly increase the removal rate of textile dyes due to the generation of sulfate radical ($SO_4^{\bullet-}$), which has an even higher redox potential of 2.5–3.1 V than $HO^\bullet$ [87,88]. UV/PS and UV/PMS processes exhibited higher removal and lower electrical energy consumption than the UV/$H_2O_2$ process in degrading certain textile dyes, such as Brilliant Green [89,90].

The main advantages of photochemical processes include ease of operation, the use of relatively inexpensive oxidants, and the absence of sludge production during the treatment. Additionally, they can disinfect waters while degrading textile pollutants [91,92]. The main limitation on full-scale photochemical application is that photochemical AOPs are easily influenced by the turbidity and inorganic anions in the water matrices due to their low UV transmittance and/or strong scavenging effect. As a result, photochemical AOPs are inapplicable for the treatment of highly polluted textile wastewater. In addition, quantum yield under UV irradiation is low for both $H_2O_2$ (1.0) and PS (1.4), requiring a high concentration of oxidants [93].

### 3.2.2. Influencing Factors

The effectiveness of photochemical AOPs is dependent on a number of operating parameters, including UV light intensity, oxidant concentration, pH, and water matrix. UV irradiation intensity and wavelength have significant effects on the removal rate of textile dyes. UV irradiation sources are typically low-pressure (LP) and medium-pressure (MP) mercury lamps with monochromatic or polychromatic emission spectra, respectively. Low-pressure UV-C light has been the most frequently used UV source.

When the oxidant concentration is low, it can facilitate the reaction rate and have an inhibitory effect when it exceeds the optimal values. The amount of radiation flux absorption capacities of the oxidant and textile dyes should be calculated prior to treatment to avoid excessive chemical addition [29]. For example, $H_2O_2$ concentration is a rate-limiting factor in the $UV/H_2O_2$ process at low doses. However, excessive $H_2O_2$ dosing can lead to the self-scavenging of $OH^\bullet$ and produce $HO_2^\bullet$ with a lower redox potential. Aleboyeh et al. reported that the optimal operating condition is at $[H_2O_2]_0/[AO7]_0 = 30$ [28]. UV irradiation of 25 and 120 min can achieve nearly 100% decolorization and 95% total organic carbon (TOC) removal of 17.5 mg $L^{-1}$ AO7 dye, respectively. The optimal $H_2O_2$ concentration in $UV/H_2O_2$ for effluent decolorization was 50 times greater than that in the $O_3/H_2O_2$ process.

Inorganic anions, such as $Cl^-$ $SO_4^{2-}$, $NO_3^-$, and $HCO_3^-$, can partially inhibit the treatment efficiency of photochemical AOPs, due to their lower reactivity with radicals. Humic acid (HA) may also significantly impair the treatment efficiency of photochemical AOPs, due to the strong competition between natural organic matter (NOM) and radicals. Rehman et al. found that the addition of (HA) and inorganic anions inhibited the degradation of Brilliant Green by UV/PS in the following order: $NO_2^- > HA > HCO_3^- > Cl^- > NO_3^- \approx SO_4^{2-}$ [89]. Additional recent publications conducted on the textile wastewater treatment using photochemical AOPs are listed in Table S3.

### 3.3. Photocatalysis

### 3.3.1. Description

Heterogeneous photocatalytic reactions involve the use of multiple irradiating solid photocatalysts to generate free radicals. Photocatalysts are easily photo-excited, resulting in the formation of electron-donating and electron-accepting sites, which induce photo-excitation reactions. Photo-excitation reactions occur when the absorbed UV photons have an energy (hv) equal to or greater than the semiconductor energy gap (between the valence and the conducting bands). Electrons ($e^-$) and holes ($h^+$) pairs are produced in the conduction and valence bands, which can either recombine or migrate to the semiconductor (S) surface and then react with chemical species adsorbed on the surface. A subsequent series of water ionization, oxygen ionosorption, and superoxide protonation reactions occur to generate $HO^\bullet$ [94,95].

The main advantages of photocatalysis for textile dye removal include: (1) they can be operated at ambient and mild conditions; (2) photochemically stable catalysts are commercially and easily available; and (3) the treatment process can potentially use sunlight as a clean and economical photo-source [1]. The disadvantages of photocatalysis for textile dyes removal include: (1) inefficient light utilization and low quantum efficiency of $HO^\bullet$ generation; (2) mass transfer limitations to the surface of the immobilized catalyst on a substrate; and (3) high recombination rate for the photoelectron and holes pairs [35].

### 3.3.2. Influencing Factors

Photocatalysis effectiveness is largely dependent on the catalyst type, catalyst form, and irradiation source. Numerous semiconductor materials have been investigated as photocatalysts, including $TiO_2$, ZnO, and $WO_3$. $TiO_2$ is the most frequently used photocatalyst due to its chemical and thermal stability, strong mechanical properties, super hydrophilicity, and low excitation energy (3.2 eV). Its low cost and non-toxicity have contributed to its widespread application [1]. Doped composites have also been extensively

investigated to increase the active surface area and photocatalytic activity of $TiO_2$, such as $TiO_2/ZnO$, $TiO_2/SiO_2$, and $SrTiO_3/CeO_2$, etc. [68,96,97]. Arcanjo et al. investigated the photocatalytic treatment of biologically treated textile mill wastewater using the $TiO_2$ modified with hydrotalcite and iron oxide ($HT/Fe/TiO_2$) under UV-vis irradiation [98]. The novel composite $HT/Fe/TiO_2$ removed more color (96%) than unmodified $TiO_2$ (88%) at pH 10 and a dose of $2 \ g \ L^{-1}$.

The shape of catalysts also influences the treatment efficiency of photocatalysis. Catalyst can be used in dispersed form (particles suspended in liquid) or thin-film form (immobilized photocatalysts onto a supporting solid matrix in a reactor). While dispersed catalysts are convenient to use and can be aerated to avoid electron-hole pair recombination, the catalytic particles can lead to the generation of dark catalytic sludge, increasing the operation cost. By increasing the dose of dispersed catalyst, the reactive sites can be increased, thereby increasing the treatment efficiency of photocatalysis. However, excessive dispersed catalysts may shield the light, block the photon penetration, and, thus, lead to light energy loss. In comparison, the insoluble catalyst film does not require the separation of catalysts following treatment, but it has a high demand for chemical stability and activity of the catalytic layer.

Irradiation source and intensity have significant effects on the removal rate of textile dyes. UV light is the most frequently investigated irradiation source because UV irradiation is required to activate $TiO_2$. Conventional UV-driven photocatalysis is based on low and medium-pressure UV light due to its cost-effectiveness. UV-LED has also been investigated due to its high quantum yield and low power consumption [99]. In addition, photocatalysts can absorb more photons and generate more electron-hole pairs on the catalyst surface with higher UV intensity, increasing the concentration of $HO^{\bullet}$ and, thus, the removal rate of textile dyes. Additional recent publications on photocatalytic AOPs for textile wastewater treatment are listed in Table S4.

### 3.4. Fenton

### 3.4.1. Description

The Fenton process was first developed by H.J.H. Fenton while investigating the destruction of tartaric acid in the late 19th century [100], and it became one of the most ubiquitous AOPs for the removal of organic pollutants in real wastewaters and soils in the 20th century [101]. The classical Fenton process generates a large amount of $HO^{\bullet}$ when $Fe^{2+}$ and $H_2O_2$ are combined under acidic conditions. $Fe^{2+}$ readily oxidizes to $Fe^{3+}$ in a matter of minutes in the presence of excess $H_2O_2$. The reaction propagates rapidly due to the regeneration of the $Fe^{2+}$ obtained from the reduction of $Fe^{3+}$.

Photo-Fenton and electro-Fenton processes are even more efficient at decolorizing and mineralizing textile dyes than the classical Fenton process. The classical Fenton reaction can be enhanced by photo-assisted irradiations, such as UV light and solar light [102]. A combination of photo-irradiation and $H_2O_2$ with $Fe^{2+}$ or $Fe^{3+}$ can be called photo-Fenton, which generates more $HO^{\bullet}$ compared to the classical Fenton reaction [103]. The direct decomposition of $H_2O_2$ molecules by UV light leads to the generation of additional $HO^{\bullet}$. In the meantime, $Fe^{2+}$ can be regenerated to catalyze Fenton's reaction due to the reductive photolysis of $[Fe(OH)]^{2+}$ (iron hydroxyl complex) and $Fe^{3+}$. Solar photo-Fenton has been developed by using free and renewable sunlight as the irradiation source in the photo-Fenton process for textile effluent treatment. It is an economically and environmentally friendly method [104].

The electro-Fenton process involves electrochemical modifications in the Fenton process. It is also one of the most widely used indirect electrochemical techniques over the last decade [105]. The electro-Fenton process can generate $HO^{\bullet}$ via the simultaneous electrogeneration of $H_2O_2$ and electroregeneration of $Fe^{2+}$ at the cathode. $H_2O_2$ is produced in situ in the electrolytic medium by supplying dissolved $O_2$ at the cathode surface in acidic conditions. $Fe^{3+}$ is then regenerated to $Fe^{2+}$ by cathodic reduction. Badawy and Ali investigated the effectiveness of Fenton oxidation and coagulation processes for the treat-

ment of industrial wastewater generated by textile companies in Egypt [106]. They found that coagulation-flocculation was ineffective against refractory and non-biodegradable dyestuffs, whereas the Fenton process removed up to 100% color and over 90% COD. This final effluent was reported to have met the requirement of the Egyptian Environmental law for water reuse.

The main advantage of the classical Fenton process is that it requires no external energy and uses relatively inexpensive chemicals. Thus, it can effectively remove recalcitrant organic dyes from textile wastewater at a low cost [107]. In addition, the Fenton process occurs at room temperature and atmospheric pressure, making it an easy and flexible process to implement in full-scale plants [11]. The major disadvantage of Fenton-based AOPs is their narrow pH range. Industrial textile wastewater is typically a basic solution, whereas the classical Fenton process is limited to an acid pH range of 2–4 [108]. Additional chemicals and manpower are required to acidify the raw water and then neutralize the treated water before disposal. High concentrations of $Fe^{2+}$ are usually used to ensure the effective treatment of textile wastewater. By adding ferrous salt, a large amount of iron-loaded sludge can be produced, which requires further treatment and proper disposal at the end. Unused $Fe^{2+}$ is retained in the treated effluent. The leftover unused $Fe^{2+}$ could substantially increase the iron levels in the effluent to be discharged into the environment and exceed the threshold. The removal of residual iron from the effluent will increase operating costs.

### 3.4.2. Influencing Factors

The efficiency of Fenton-based AOPs is dependent on various factors, such as pH and the dose of Fenton's reagent. The solution pH is critical for Fenton-based AOPs since it controls the rate of $HO^\bullet$ generation and the coexistence of dissolved $Fe^{2+}$ and $Fe^{3+}$. The classical Fenton process requires a low pH value of around 3. When the solution pH increases, the concentration of the $Fe^{3+}$ species decreases due to the precipitation of $Fe(OH)_3$. When the solution pH decreases, the formation of $Fe^{2+}$ complexes occurs, resulting in a decrease in the $Fe^{2+}$ concentration. Patel et al. used the Fenton process to decompose Reactive Red 241 at different pHs (4, 7, and 10) [30]. They reported that the decolorization of Reactive Red 241 was 28.5% at pH 4, while the Fenton process showed insignificant decolorization results at neutral and basic pH values (less than 10%). Hayat et al. investigated the efficiency of the Fenton process with and without pH adjustment in decolorizing and mineralizing dye in real textile wastewater [20]. The results show that Fenton's reagent is the most effective for color and COD removal at pH 3. Buthiyappan and Abdul Raman found that the optimal pH value for photo-Fenton treatment of recalcitrant industrial wastewater is around 5.36 [42]. The concentration of Fenton's reagent and the ratio of $H_2O_2/Fe^{2+}$ significantly influences the efficiency of Fenton-based AOPs. Insufficient Fenton reagent concentration leads to a lack of $HO^\bullet$ and a decrease in reaction efficiency. However, the overdose of $H_2O_2$ may scavenge $HO^\bullet$ and inhibit the degradation of pollutants, whereas the overdose of $Fe^{2+}$ may lead to increased sludge generation. Moreover, excessive concentration of Fenton reagent will increase costs and limit practical application. Wang and Zhuan reported that the theoretically optimal $H_2O_2$ concentration can be calculated based on the chemical equation [78]:

$$C_aH_bN_cO_d + (2a + 0.5b + 2.5c - d)H_2O_2 \rightarrow aCO_2 + (2a + b + 2c - d)H_2O + cHNO_2 \quad (1)$$

Namely, one mole of $C_aH_bN_cO_d$ requires $(2a + 0.5 b + 2.5c - d)H_2O_2$ moles $H_2O_2$, with the actual required $H_2O_2$ concentration always being slightly higher than the calculated value. Paździor et al. reported that the optimal Fenton reagent concentration for treating real industrial wastewater is approximately 0.8 g $H_2O_2$ and 0.06 g $Fe^{2+}$ per 1 g initial COD [4]. Ribeiro et al. achieved a COD removal of 66% when the mass ratio of $[COD]/[H_2O_2]/[Fe^{2+}]$ = 1:2:2 was used to treat textile wastewater [109]. Buthiyappan and Abdul Raman found that the optimal mass ratios for photo-Fenton treatment of recalcitrant industrial wastewater are around $[H_2O_2]/[COD]$ = 8.87 and $[H_2O_2]/[Fe^{2+}]$ = 4.82 [42].

Yu et al. investigated the feasibility of using online monitoring of dissolved oxygen and oxidation- reduction potential as critical parameters for controlling the electro-Fenton process used to treat textile wastewater [110]. They found that both parameters exhibit strong correlations with variations in $H_2O_2$, $Fe^{2+}$, and $Fe^{3+}$, which may contribute to identifying $H_2O_2$ overdosage and reducing chemical costs.

Textile wastewater contains complex matrices, such as $Cl^-$, $HCO_3^-$, $CO_3^{2-}$, that can act as $HO^\bullet$ scavengers. Additional operating parameters, such as UV intensity (for photo-Fenton), applied current, and supporting electrolyte (for electro-Fenton), can also influence the effectiveness of Fenton [111]. Additional recent publications on the textile wastewater treatment using Fenton-based AOPs are listed in Table S5.

### 3.5. Electrochemical

#### 3.5.1. Description

Electrochemical AOPs (EAOPs) can produce reactive species $HO^\bullet$ and, thus, oxidize and decompose organic compounds into non-toxic compounds when an electric current is applied [112].

EAOPs utilize two distinct electron transfer mechanisms, i.e., direct oxidation and indirect oxidation. Direct oxidation occurs at anode (M) surface where the pollutants are adsorbed. Organic pollutants can lose electrons to generate small molecular products [113]. Indirect oxidation occurs in the liquid electrolyte that is mediated by oxidants. Anions in water react with the anode to produce reactive oxygen species in situ [31].

Among the various EAOPs, anodic oxidation is the most frequently studied since it uses the most direct and environmentally friendly method of electrochemically producing $HO^\bullet$. Anodic oxidation of water can directly produce $HO^\bullet$ at the surface of anode material (M) with high $O_2$ evolution overvoltage anodes [114]. Heterogeneous reactive oxygen species absorbed on the anode material ($M(HO^\bullet)$) can subsequently oxidize organic matters (R) to produce $CO_2$, $H_2O$, and inorganic ions (X).

The main advantage of EAOPs is that they are clean and environmentally compatible since the main reagent is an electron. In addition, EAOPs are simple to control, have the potential for automation, require no chemicals, and produce no sludge during treatment [114,115]. The disadvantages of EAOPs include their low mass transfer efficiency and high electricity consumption. Furthermore, hazardous material can be generated on the anode surface at low potentials, posing a poisoning risk from EAOPs [105].

#### 3.5.2. Influencing Factors

Anode materials and electrolytes are critical parameters that significantly influence the efficiency of EAOPs at removing pollutants. Initial choices for anode materials for anodic oxidation, as reported in other studies, include Pt [105], $PbO_2$ [37], $SnO_2$ [116], $IrO_2$, $RuO_2$, Ti/Ru alloy [40], and Ti/Pt alloy [31,47,53,117–120]. Improved anodes, such as boron-doped diamonds (BDD), appear to be promising for EAOP treatment of textile wastewaters [121]. Compared to conventional anodes, BDD exhibits superior chemical and electrochemical stability, a broad electrochemical operating range, high oxidation power, and relatively low production costs. In addition, BDD has a significantly higher $O_2$ overvoltage than other anodes, which results in a greater amount of $HO^\bullet$ physisorbed on the anode surface [122]. Treatments of real textile effluent confirm that the color and COD can be completely removed by EAOP with BDD anode [37,123–125]. Dimensionally stable anodes (DSA) are another alternative electrode material for textile wastewater treatment due to their chemical stability, low cost, and long lifetime [116,126]. The most commonly used DSA is the $Ti/Ru_{0.3}Ti_{0.7}O_2$ ($Cl_2$-evolution) anode [32,127]. However, high efficiency can be achieved only when active $Cl_2$ is used as a catalyst in indirect oxidation [38].

The efficiency of anodes can be improved by adding suitable substrates. Metal oxides and DSA anodes can facilitate the generation of $Cl_2$ from aqueous $Cl^-$, thus increasing oxidation efficiency through the generation of the relatively strong oxidant HClO. Chatzisymeon et al. investigated the EAOP treatment of textile dyes and dyehouse ef-

fluents [31]. They reported that the decolorization efficiency can only reach around 60% after 180 min of treatment using real wastewater without NaCl. However, when 0.5% NaCl was added to the effluent, color removal increased to over 95% within 10–15 min of treatment. Malpass et al. reported that the addition of 0.1 g L$^{-1}$ NaCl to the effluent may cause the generation of $Cl_2$ at the anode, leading to a current increase at potentials over 1.2 V [32]. Apart from HClO, chlorine-mediated oxidation can be also used to generate oxidants, such as $O_3$ and $S_2O_8{}^{2-}$ in situ. Martínez-Huitle et al. reported that the COD removal efficiency of EAOP can be significantly improved when $Na_2SO_4$ was added into an electrolyte using BDD as the electrode [119]. The reason is that the electrolysis BDD anodes in electrolytes containing $SO_4{}^{2-}$ may favor the electrogeneration of $S_2O_8{}^{2-}$ on the BDD surface. Baddouh et al. evaluated the decolorization efficiency of Rhodamine B dye using two anions ($Na_2SO_4$ and NaCl) as supporting electrolytes [116]. They found that adding NaCl enhanced the oxidation capacity of DSA electrodes more effectively than $Na_2SO_4$ alone. Color removal increased from 19.6% in the $Na_2SO_4$ electrolyte to nearly 100% in the NaCl electrolyte after 1 h of EAOP treatment. Similar results were observed on the $SnO_2$ electrode, with color removal increasing from approximately 14.3 to 90.3% with the addition of $Na_2SO_4$ and NaCl, respectively. Solano et al. confirmed that dissolved salts (NaCl and $Na_2SO_4$) have a significant influence on the removal of color and COD from real textile industrial effluent [128]. The electrogeneration of strong oxidant species ($Cl_2$ and $S_2O_8{}^{2-}$) on the BDD surface can rapidly attack the chromophore group of textile dyes, allowing for a rapid color removal.

In addition, the removal efficiency of pollutants is also strongly affected by the applied current density and the initial pollutant concentration. Increased applied current densities combined with lower pollutant concentrations can result in an increase in removal efficiency. Additional recent publications on textile wastewater treatment using EAOPs are listed in Table S6.

*3.6. Ultrasound*

3.6.1. Description

Ultrasound (US), also known as sonolysis, was initially applied for degassing water and accelerating chemical reactions [129]. It is based on the cavitation phenomenon, in which bubbles containing dissolved gases and water vapor are generated. The pyrolysis of water at US frequencies of 20–500 kHz can result in the collapse of micro-bubbles due to acoustical wave-induced compression and rarefaction. HO$^\bullet$ can be produced due to the high intensity of acoustic cavity bubbles [36]. Pollutants can be oxidized by pyrolysis of ultrasound or the production of HO$^\bullet$.

The coupling of US with oxidants (sono-chemical), UV irradiation (sono-photolysis), catalysts (sono-catalysis), and Fenton (sono-Fenton) has received increased attention. Among these, the US/$O_3$ is the most frequently investigated approach. The combination of US irradiation with dissolved $O_3$ leads to the thermal decomposition of $O_3(g)$ in the cavitation bubbles [130]. The generated ground-state O atoms may have synergistic effects, resulting in the generation of additional HO$^\bullet$.

The main advantage of the US process is that no additional chemicals are required, which results in the absence of sludge during treatment. US process can also improve homogenization, facilitate solubilization, assist mass transfer, and shorten reaction time when coupled with chemical reactions [34]. The main disadvantage of US process is that the degradation rate for an organic pollutant is extremely low, making it energy-intensive. In addition, a large amount of dissolved oxygen is required for the process [131].

3.6.2. Influencing Factors

The performance of US-based AOPs depends on the parameters influencing cavitation and bubble collapse, such as dye concentration, power density, mechanical agitation, temperature, reaction medium properties (vapor pressure, viscosity, surface tension),

sound wave frequency and intensity, and gas properties (solubility, specific heat, thermal conductivity), etc. [33,34].

The US frequency influences acoustic cavitation and sonochemistry. The optimal frequency is determined by the characteristics of the dye and its localization within or on the surface of the cavitation bubbles. Dede et al. investigated the ultrasonic degradation of C.I. Reactive Orange 107 at three frequencies (378, 850, and 992 kHz) using a dye dose of 50 mg $L^{-1}$ and temperature of 20 °C [34]. They reported that the decolorization rate increased up to 99% with increasing frequency from 376 to 850 kHz, but decreased dramatically to 69% when the frequency value further increased to 992 kHz. The results showed that the optimal frequency for dye removal was 850 kHz. Increased dye concentration can lead to a decrease in decolorization efficiency due to insufficient $HO^{\bullet}$ generation and diminished cavitation effect at higher dye loads. Additional recent publications on the textile wastewater treatment by US-based AOPs are listed in Table S7.

## 4. Cost Comparison

To make a direct cost comparison of AOPs, the figure of merit of AOPs has been developed by Bolton and accepted by the International Union of Pure and Applied Chemistry [132]. It is defined as "electrical energy per order" ($E_{EO}$, kWh $m^{-3}$ order$^{-1}$), which refers to the electrical energy (kWh) required to degrade a compound by one order of magnitude in 1 $m^3$ of contaminated water regardless of the nature of the system [93,133].

$$E_{EO} = \frac{P \times t \times 1000}{V \times \log\left(\frac{C_i}{C_f}\right)} \tag{2}$$

where P is the power (kW) used during AOP, t is the treatment time (h), V is the treated volume (L), $C_i$ and $C_f$ are the initial and final concentrations of the pollutant (mg $L^{-1}$).

Before putting an AOP into industrial use, it is necessary to consider both technical feasibility and economic cost. Electrical energy per order ($E_{EO}$) is a useful figure of merit for directly assessing and comparing AOPs based on economic energy cost. $E_{EO}$ values of ozonation, photochemical, photocatalysis, Fenton, electrochemical, and ultrasound under laboratory experimental conditions were calculated using data extracted from peer-reviewed publications (27, 70, 50, 34, 42, and 18 publications, respectively, for each AOP technique, providing 241 datapoints). The detailed $E_{EO}$ results for each AOP are presented in Tables S2–S7. It should be noted that $E_{EO}$ values are highly dependent on the physicochemical characteristics of the pollutant, pH, oxidant concentration, etc. [134]. Therefore, $E_{EO}$ values of a particular AOP can vary by several orders of magnitude. Despite the wide range of possible values due to various laboratory conditions, noticeable differences in $E_{EO}$ values can still be observed.

Figure 4 shows the comparison of $E_{EO}$ values of various AOPs for textile wastewater treatment derived from the 241 datapoints. Fenton-based AOPs show the lowest median $E_{EO}$ values of 0.98 kWh $m^{-3}$ order$^{-1}$, followed by photochemical (3.20 kWh $m^{-3}$ order$^{-1}$), ozonation (3.34 kWh $m^{-3}$ order$^{-1}$), EAOP (29.5 kWh $m^{-3}$ order$^{-1}$), photocatalysis (91 kWh $m^{-3}$ order$^{-1}$), and US (971.45 kWh $m^{-3}$ order$^{-1}$). The results show that US is the most energy-intensive process and Fenton is the most cost-effective method for textile wastewater treatment. Miklos et al. reported that the median $E_{EO}$ values of various AOPs followed the order of ozonation < UV/chemical < EAOP < UV/catalyst < ultrasound, which is consistent with the findings in this study [35].

The Fenton process can degrade textile wastewater at the lowest costs, due to the relatively inexpensive Fenton's reagent and low energy input requirements. As a result, it shows great potential for full-scale application [11]. Heterogeneous photocatalysis processes have higher operating costs than homogeneous photochemical and ozonation processes for textile wastewater treatment. The reason may be due to the low mass transfer to the surface of the catalyst on a substrate and the low quantum yield of photocatalytic reactions for $OH^{\bullet}$ production. As a result, currently, the heterogeneous process is rarely

applied in industrial textile water treatment plants, despite extensive research efforts [35]. Durán et al. also concluded that the homogeneous photo-Fenton process is significantly more cost-effective than heterogeneous photo-catalysis for pharmaceutical treatment in water, particularly when ferrioxalate is used as the catalyst [135]. Due to their highly energy consumption, US-based processes have the lowest electrical efficiency among all the AOPs. The results indicate that while US as a standalone process may not yet be economically feasible for textile wastewater treatment, it may be applied as an auxiliary tool with other AOPs such as UV or ozonation [36]. It can also be used in conjunction with biological treatment as a final step to increase the treatment efficiency [34].

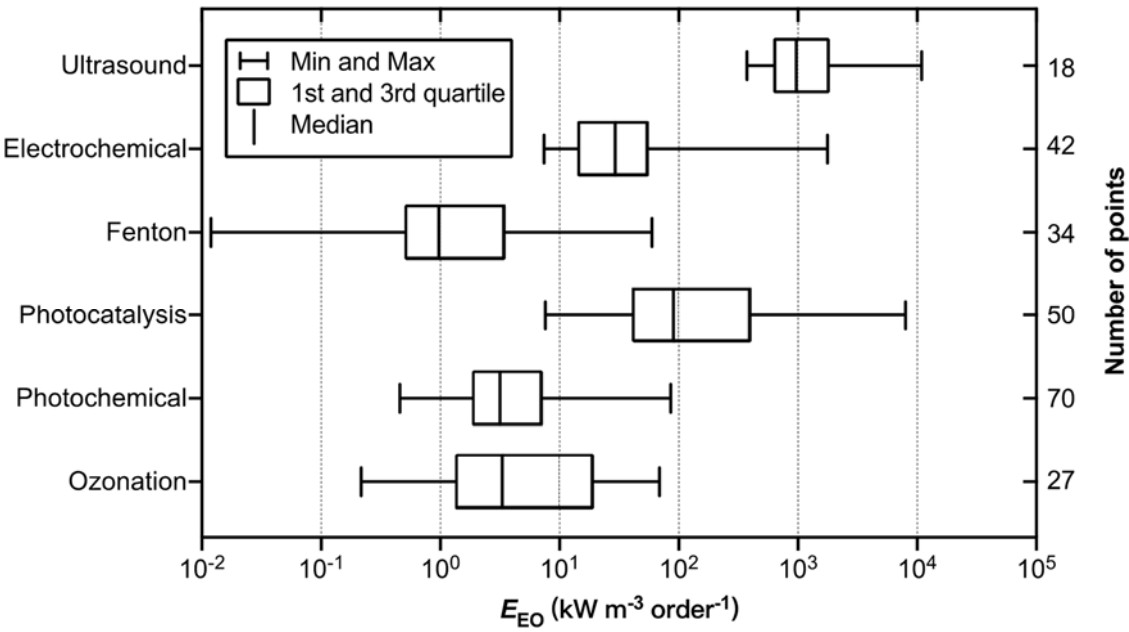

**Figure 4.** $E_{EO}$ values of various AOPs on textile wastewater treatment.

It is worth noting that high removal efficiency does not necessarily represent low energy consumption. For example, Azbar et al. compared the COD and color removal efficiency of various AOPs (including $O_3$, $UV/H_2O_2$, $UV/O_3$, $UV/H_2O_2/O_3$, and Fenton) from polyester and acetate fiber dyeing effluents [136]. They reported that for 90% color removal, $O_3$, $UV/H_2O_2$, $UV/O_3$, $UV/H_2O_2/O_3$, and Fenton processes required operating costs of 5.28, 1.26, 6.38, 6.54, and 0.23 \$ m$^{-3}$, respectively, while their COD removals are 92%, 90%, 94%, 99%, and 96%, respectively. Although $UV/H_2O_2/O_3$ removed the most COD, the Fenton process was the most cost-effective.

## 5. Full-Scale Application in a Textile WWTP

The application of AOP to a textile WWTP, operated by Sateri (a viscose fiber producer), is presented as an illustrative example of moving from bench-scaled laboratory studies to full-scale implementation. Sateri's viscose staple fiber mill has been operating in Jiangxi Province (China) since 2004, with an AOP-based WWTP commissioned in 2018. This WWTP is reported to operate continuously, with a treatment capacity of 47,000 m$^3$ d$^{-1}$. As part of the treatment process, a combined cyclic activated sludge system followed by a fluidized-bed Fenton post-treatment process has been established (flow diagram shown in Figure 5). Sand quartz is utilized as carrier during the treatment to increase the treatment efficiency and reduce the operating costs. The initial biological treatment of raw effluent has been reported to reduce COD to 60–80 mg L$^{-1}$ and color to 30–70 ADMI. The fluidized-bed Fenton process subsequently reports a reduction in the COD to 20–40 mg L$^{-1}$ and color to 8–10 ADMI. Based on these estimates, the overall removal percentages of colorization and COD would be 85% and 90%, respectively, which would indicate that the fluidized-bed

Fenton process has been able to treat bio-treated textile wastewater more efficiently and at an industrial operation scale. The effluent discharge from this treatment plant has reported to achieve the Grade I-A discharge standard of pollutants for municipal WWTP GB 18918-2002. While these reported removal percentages are plausible when compared against the reviewed literature from laboratory bench-studies (Section 2), the short time-frame of the operation of the WWTP with its high daily load would require continuous and diligent monitoring to build further confidence in the application.

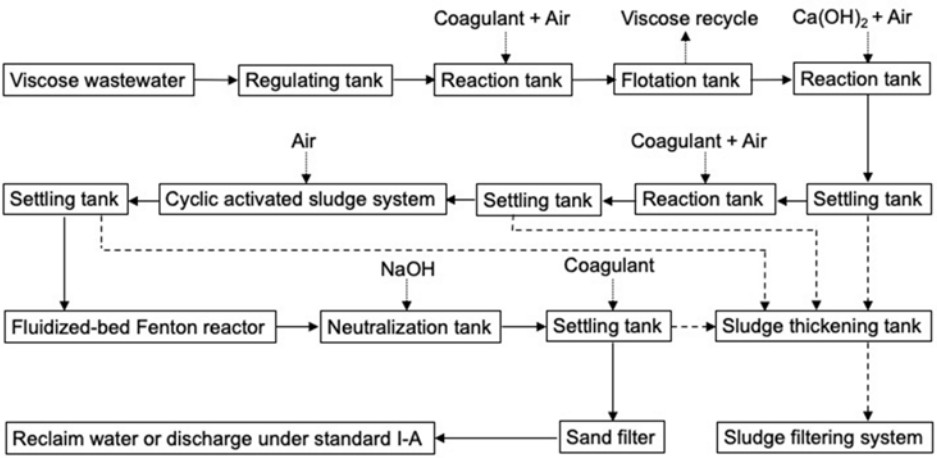

**Figure 5.** Flow diagram of Sateri viscose fiber mill WWTP.

The operating costs of the process is estimated to be CNY ¥1.62 m$^{-3}$ (USD $0.23 m$^{-3}$), which corresponds to approximately 3.24 kWh m$^{-3}$, according to the electricity tariff in China (CNY ¥0.5 kWh$^{-1}$). High turbidity and high concentration of pollutants in the raw textile wastewater may act as radical scavengers, reducing treatment efficiency and, thus, increasing the actual operating costs.

## 6. Conclusions and Future Directions

Untreated or inadequately treated textile wastewater has extremely high COD of 413–8000 mg L$^{-1}$, BOD$_5$ of 160–491 mg L$^{-1}$, suspended solids of 62–2545 mg L$^{-1}$, ADMI unit color of 625–2175, and a basic solution pH—representing a high burden on downstream freshwater ecosystems, and a risk for dependent human populations. AOPs are generally more effective than conventional oxidation processes due to the generation of highly reactive radicals, thus, they constitute alternative technologies for the remediation of textile wastewater. The presence of inorganic and organic water constituents often inhibits the treatment of target pollutants via radical scavenging mechanisms, which leads to a lower treatment effect. As a result, the water quality parameters (and local regulatory requirements) must be taken into consideration. The median E$_{EO}$ values of various AOPs followed the order of Fenton (0.98 kWh m$^{-3}$ order$^{-1}$), photochemical (3.20 kWh m$^{-3}$ order$^{-1}$), ozonation (3.34 kWh m$^{-3}$ order$^{-1}$), EAOP (29.5 kWh m$^{-3}$ order$^{-1}$), photocatalysis (91 kWh m$^{-3}$ order$^{-1}$), and US (971.45 kWh m$^{-3}$ order$^{-1}$), indicating that the Fenton process is the most cost-effective method to remove textile waste based on the energy requirement analysis. A full-scale application of a textile WWTP is introduced as an illustrative example for scaling of the technology. Results from this application, as reported by Sateri, indicate that enhanced Fenton technology can treat textile effluent at an industrial scale (47,000 m$^3$ d$^{-1}$) at a cost of CNY ¥1.62 m$^{-3}$ (USD $0.23 m$^{-3}$), with colorization and COD removal of 85% and 90%, respectively.

Many laboratory experiments have been done over the past two decades concerning the AOP treatment kinetics of textile wastewater, indicating an increasing research interest for these innovative approaches. More investigation on removal mechanisms and the ecotoxicological safety of intermediate products, and analysis of adoption at a real

industrial-scale for treating textile effluents are encouraged for future studies. Additional efforts should be made to fabricate metal-organic frameworks (MOFs)-based catalysts, such as Fe(II) substitution and melamine foams, to enhance the treatment performance of AOPs and facilitate their engineering application [137–139].

Conventional processes can also be combined with AOPs to achieve better performance with a lower energy requirement. Economic models are imperative to be developed to assess the energy cost of textile wastewater treatment using combined AOPs.

**Supplementary Materials:** The following are available online at https://www.mdpi.com/article/10.3390/w13243515/s1, Table S1: Oxidation potential of various oxidizing agents. Table S2: Textile wastewater treatment using O3-based AOPs. Table S3: Textile wastewater treatment using photochemical AOPs. Table S4: Textile wastewater treatment using photocatalytic AOPs. Table S5: Textile wastewater treatment using Fenton-based AOPs. Table S6: Textile wastewater treatment using electrochemical AOPs. Table S7: Textile wastewater treatment using US-based AOPs.

**Author Contributions:** Y.Z. conceived the original idea, performed the analytic calculations, and took the lead in writing the manuscript; K.S. and D.V. reviewed the manuscript, provided critical feedback, aided in interpreting the results, and helped shape the manuscript; and C.M. provided the details of the full-scale implementation example. All authors have read and agreed to the published version of the manuscript.

**Funding:** This work was carried out under a joint CI-Sateri project and funded by a grant from Sateri.

**Conflicts of Interest:** The authors declare no conflict of interest.

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
