# Peer review of "Treatment of Textile Wastewater Using Advanced Oxidation Processes—A Critical Review"

_water, doi:10.3390/w13243515_

Round 1

Reviewer 1 Report

Excellent review, with a wealth of very well organized information.

Author Response

Thanks for recognizing our effort. We really appreciate your kind comments.

Reviewer 2 Report

In this review article the authors propose a summary of the main results of the studies on the degradation of textile wastewater by advanced oxidation processes, mainly published in the last twenty years. The article is basically well written, however I have some suggestions that I believe the authors need to think about.
1. Most of the articles reviewed have been published from the first years of this century to today, however there are also some articles from the last century. This type of review should focus on recent articles, set a time limit which should be at most 5 years. This aspect is very important because by measuring the number of articles per topic, in recent years, it provides a precise evaluation of the interest in the topic. If not enough articles have been found in the last 5 years to build a review, the topic may be of little interest. The authors should reflect on this.
2. The abstract is not balanced, typically there is a background, methods and main results. This abstract is poor in results.
3. The article reads well, however I encourage the authors to add some figures to the text, especially in the first paragraphs, to show some trend and / or mechanism that helps the reader to focus on the topic.
4. "Critical review". In the text of the review there are indeed numerous considerations and critical evaluations, however the conclusions are too schematic and just sufficient.
5. In the tables, the significant figures must be consistent.

Author Response

Thanks for your kind review. The point-by-point response to your comments are provided as below.

1. Most of the articles reviewed have been published from the first years of this century to today, however there are also some articles from the last century. This type of review should focus on recent articles, set a time limit which should be at most 5 years. This aspect is very important because by measuring the number of articles per topic, in recent years, it provides a precise evaluation of the interest in the topic. If not enough articles have been found in the last 5 years to build a review, the topic may be of little interest. The authors should reflect on this.

Response: Thanks for your insightful perspective. We agree that the review that focus on recent 5 years will provide the most precise and updated evaluation of the topic. However, electrical energy per order (EEO), the figure of merit of AOPs, has been developed by Bolton and accepted by the International Union of Pure and Applied Chemistry on 2001 (Bolton, etc. Figures-of-merit for the technical development and application of advanced oxidation technologies for both electric- and solar-driven systems. Pure Appl. Chem. 2001, 73, 627–637). The publications on the cost assessment of AOPs for textile wastewater treatment started from the first years of this century, as shown in Figure 2 in the revised manuscript. We want to make a thorough comparison of electrical energy per order (EEO) values of various AOPs, using all the available datapoints since the creation of the assessment method. We hope that explains why we reviewed the articles published from the first years of this century to today.

2. The abstract is not balanced, typically there is a background, methods and main results. This abstract is poor in results.

Response: Thanks for your suggestion. We have added more main findings and conclusions to the abstract to make it more balanced. Please refer to the revised abstract.

3. The article reads well, however I encourage the authors to add some figures to the text, especially in the first paragraphs, to show some trend and / or mechanism that helps the reader to focus on the topic.

Response: Thanks for your suggestion. We have included three additional figures to section 1 and 2. The first schematic diagram illustrates the advanced oxidation processes (AOPs) used in textile wastewater treatment, allowing readers to have a better understanding of the AOPs types and the paper’s structure. The second figure displays the number of publications on the cost assessment of AOPs for textile wastewater treatment during the last two decades, indicating the topic's popularity. The third figure shows the significant gap between the pollutant concentration in raw textile wastewater and discharge standards of major pollutants.

4. "Critical review". In the text of the review, there are indeed numerous considerations and critical evaluations, however, the conclusions are too schematic and just sufficient.

Response: Thanks for the suggestion. We’ve added additional information to the conclusions to make it more sufficient. Please refer to the conclusion in the revised manuscript.

5. In the tables, the significant figures must be consistent.

Response: The inconsistency of the significant figures in the tables is because that they were derived from different references. We suggest retaining the current significant figures to keep the accuracy.

Reviewer 3 Report

This manuscript summarized and analyzed the advanced oxidation processes (AOPs) for the treatment of textile wastewaters. I would recommend a major revision before its publication in this journal.

  1. One of the most concerns is that "degradation" should be used to describe a certain organic pollutant, such as dyes and antibiotics, rather than wastewater. It is better to use "treatment" or "purification".
  2. The mechanisms of each AOPs are missing.
  3. The catalysts for Fenton oxidation and catalytic ozonation are too conventional to be widely interested. Here are some advised advanced catalysts for applying efficient AOPs: Nano Research, DOI: 10.1007/s12274-021-3918-6, http://www.thenanoresearch.com/upload/justPDF/3918.pdf; Applied Catalysis B: Environmental, 2021, 286, 119859; Chemical Engineering Journal, 2021, 404, 127075.
  4. There are many typo and syntax errors that the authors should be carefully checked.

Author Response

Thanks for your kind review. The point-by-point response to your comments is provided as below.

1. One of the most concerns is that "degradation" should be used to describe a certain organic pollutant, such as dyes and antibiotics, rather than wastewater. It is better to use "treatment" or "purification".

Response: Thanks for your suggestion. We’ve changed all the improper use of “degradation” to “treatment” in the revised manuscript.

2. The mechanisms of each AOPs are missing.

Response: The mechanisms of each AOPs are demonstrated in Table 2.

3. The catalysts for Fenton oxidation and catalytic ozonation are too conventional to be widely interested. Here are some advised advanced catalysts for applying efficient AOPs: Nano Research, DOI: 10.1007/s12274-021-3918-6, http://www.thenanoresearch.com/upload/justPDF/3918.pdf; Applied Catalysis B: Environmental, 2021, 286, 119859; Chemical Engineering Journal, 2021, 404, 127075.

Response: Thanks for the suggestion. We have reviewed the three articles and incorporated them into our revised paper.

4. There are many typo and syntax errors that the authors should be carefully checked.

Response: Thanks for pointing out the issue. We've gone over the entire paper carefully and fixed the typos and syntax errors thoroughly.

Round 2

Reviewer 3 Report

Great work! I have no more comments.